# The effect of antidepressants on the severity of COVID-19 in hospitalized patients: A systematic review and meta-analysis

Hosein Nakhaee[1][☯], Moein Zangiabadian[2][☯]*, Reza Bayati[1][‡], Mohammad Rahmanian[1][‡], Amir Ghaffari Jolfayi[1][‡], Sakineh Rakhshanderou[3]*

1 Student Research Committee, School of Medicine, Shahid Beheshti University of Medical Sciences, Tehran, Iran, 2 Department of Microbiology, School of Medicine, Shahid Beheshti University of Medical Sciences, Tehran, Iran, 3 Environmental and Occupational Hazards Control Research Center, School of Public Health, Shahid Beheshti University of Medical Sciences, Tehran, Iran

☯ These authors contributed equally to this work.
‡ RB, MR and AGJ also contributed equally to this work.
* Zangiabadian1998@gmail.com (MZ); s_rakhshanderou@sbmu.ac.ir (SR)

**Data Availability Statement:** All relevant data are within the paper and its Supporting Information files.

## Abstract

### Introduction

Clinical Depression and the subsequent low immunity is a comorbidity that can act as a risk factor for the severity of COVID-19 cases. Antidepressants such as Selective serotonin reuptake inhibitor and Serotonin-norepinephrine reuptake inhibitors are associated with immune-modulatory effects, which dismiss inflammatory responses and reduce lung tissue damage. The current systematic review and meta-analysis aims to evaluate the effect of antidepressant drugs on the prognosis and severity of COVID-19 in hospitalized patients.

### Methods

A systematic search was carried out in PubMed/Medline, EMBASE, and Scopus up to June 14, 2022. The following keywords were used: "COVID-19", "SARS-CoV-2", "2019-nCoV", "SSRI", "SNRI", "TCA", "MAOI", and "Antidepressant". A fixed or random-effect model assessed the pooled risk ratio (RR) with 95% CI. We considered $P < 0.05$ as statistically significant for publication bias. Data were analyzed by Comprehensive Meta-Analysis software, Version 2.0 (Biostat, Englewood, NJ).

### Results

Fourteen studies were included in our systematic review. Five of them were experimental with 2350, and nine of them were observational with 290,950 participants. Eight out of fourteen articles revealed the effect of antidepressants on reducing the severity of COVID-19. Selective serotonin reuptake inhibitors drugs, including Fluvoxamine, Escitalopram, Fluoxetine, and Paroxetine, and among the Serotonin-norepinephrine inhibitors medications Venlafaxine, are reasonably associated with reduced risk of intubation or death. Five studies showed no significant effect, and only one high risk of bias article showed the negative effect of antidepressants on the prognosis of Covid-19. The meta-analysis of clinical trials showed

**Funding:** The author(s) received no specific funding for this work.

**Competing interests:** The authors have declared that no competing interests exist.

that fluvoxamine could significantly decrease the severity outcomes of COVID-19 (RR: 0.763; 95% CI: 0.602–0.966, I2: 0.0)

## Findings

Most evidence supports that the use of antidepressant medications, mainly Fluvoxamine, may decrease the severity and improve the outcome in hospitalized patients with SARS-CoV-2. Some studies showed contradictory findings regarding the effects of antidepressants on the severity of COVID-19. Further clinical trials should be conducted to clarify the effects of antidepressants on the severity of COVID-19.

## Introduction

After over two years since the first case of the novel coronavirus was detected in December 2019 in Wuhan, China, 483 million infected cases and 6.13 million deaths have been reported worldwide up to January 21, 2022 [1]. Covid19 is an acute respiratory disease, which results to progressive respiratory failure and eventually leading to death. Common symptoms include fever, dry cough, myalgia, and fatigue. Severe cases develop dyspnea, hemoptysis, and acute respiratory distress syndrome, resulting in death. The severity or mortality is higher in the older population, with comorbidities such as diabetes, hypertension, cardiovascular diseases, and weaker immune functions [2]. During the COVID-19 pandemic with consequent strict lockdown measures, psychiatric patients have suffered higher-scale episodes of anxiety, depression, and stress disorders [3]. Clinical depression and the subsequent low immunity can also act as risk factors for the severity of COVID-19 cases [4]. Patients with clinical depression have lower immunity compared to those of healthy controls. A history of depression is associated with a higher risk of infection, and the increased risk remains consistent over time [5]. The suggested mechanism of action is fewer circulating $CD4^+$ T-cells and a reduction in natural killer cell cytotoxic responses of lymphocyte proliferation in elderly patients with clinical depression [6]. Treatment of Depression as underlying morbidity could decrease the risk of clinical deterioration in Covid-19 patients. On the other hand, antidepressants such as Serotonin-norepinephrine reuptake inhibitor(SNRI) and Selective serotonin reuptake inhibitors (SSRI), which are widely used in the treatment of psychiatric patients, are shown to reduce coronavirus infection rates as patients who were receiving these medications were less likely to test positive for COVID-19 [7]. Antidepressants are also associated with less severe cases as they significantly reduced the risk of intubation or death in some cohort studies [8, 9]. Several mechanisms are suggested to justify this finding. Antidepressants could be associated with declined plasma levels of inflammatory cytokines, including IL-10, TNF-α, CCL-2, and IL-6, which are related to COVID-19 severity and mortality [10, 11]. Also Some SSRI antidepressants, such as fluvoxamine which is a functional inhibitor of acid sphingomyelinase activity (FIASMA), may prevent the infection of epithelial cells with SARS-CoV-2 [12]. In this study we aim to evaluate the effect of antidepressant drugs on prognosis and severity of COVID-19 in hospitalized patients.

## Method

This study was conducted and reported in accordance with the Preferred Reporting Items for Systematic Reviews and Meta-Analyses statement [13]. The study was registered in the Systematic Review Registration: PROSPERO (pending registration ID: 313272).

## Search strategy

We searched Pubmed/Medline, Embase, and Scopus for clinical studies reporting anti-depressants' effect on reducing the severity of hospitalized patients with Covid-19, published up to June 14, 2022. We included clinical trials, cohort and case-control studies that were written in English. We used the following MeSH terms: ´´´antidepressive agents', 'antidepressive agents, second generation', 'antidepressive agents, tricyclic', 'monoamine oxidase inhibitors', 'serotonin and noradrenaline reuptake inhibitors', 'serotonin uptake inhibitors', 'COVID-19' and 'SARS-CoV-2'" (Tables S1-S3 in S3 File). Keyword searches were done with combinations of the terms "SSRI", "SNRI", "TCA", "MAOI", "Antidepressant", "2019 novel coronavirus" and "sars coronavirus 2". Lists of references of selected articles and relevant review articles were hand-searched to identify further studies.

## Study selection

The records found through database searching were merged, and the duplicates were removed using EndNote X8 (Thomson Reuters, Toronto, ON, Canada). Two reviewers independently screened the records by title/abstract and full text to exclude those unrelated to the study objectives. The lead investigator resolved any disagreements. Included studies met the following criteria: (i) patients were diagnosed with COVID-19 based on the WHO criteria; (ii) patients received anti-depressants; and (iii) outcomes (hospitalization due to COVID-19 and/or ARDS and/or need to NIV or mechanical ventilation and/or ICU admission and/or death). Conference abstracts, editorials, reviews, study protocols, molecular or experimental studies on animal models and studies focusing on infection risk were excluded.

## Data extraction

Two reviewers designed a data extraction form. These reviewers extracted data from all eligible studies, and consensus resolved differences. The following data were extracted: first author name; year of publication; study duration; type of study, countries where the research was conducted; demographics (i.e., age, sex); Detection test of COVID-19; anti-depressant type, dosage and frequency; Follow-up time; the definition of case and control; the total number of controls and cases, severity indices, mechanism of action of anti-depressants against COVID-19 and Possibility of using anti-depressants in COVID-19 treatment.

## Quality assessment

Two blinded reviewers assessed the quality of the studies using three different assessment tools (checklists): two for observational studies (case controls and cohorts) and one for experimental studies [14]. Items such as study population, measure of exposures, confounding factors, extent of outcomes, follow-up data, and statistical analysis were evaluated.

## Statistical analysis

The pooled risk ratios (RRs) with 95% CI were assessed using random or fixed-effect models. The fixed-effects model was used because of the low estimated heterogeneity of the true effect sizes. The between-study heterogeneity was assessed by Cochran's Q and the I2 statistic. I2 values of more than 50% were considered high heterogeneity [15]. Publication bias was evaluated statistically by using Egger's and Begg's tests as well as the funnel plot ($p < 0.05$ was considered indicative of statistically significant publication bias; funnel plot asymmetry also suggests bias) [16]. All analyses were performed using "Comprehensive Meta-Analysis" software, Version 2.0 (Biostat, Englewood, NJ).

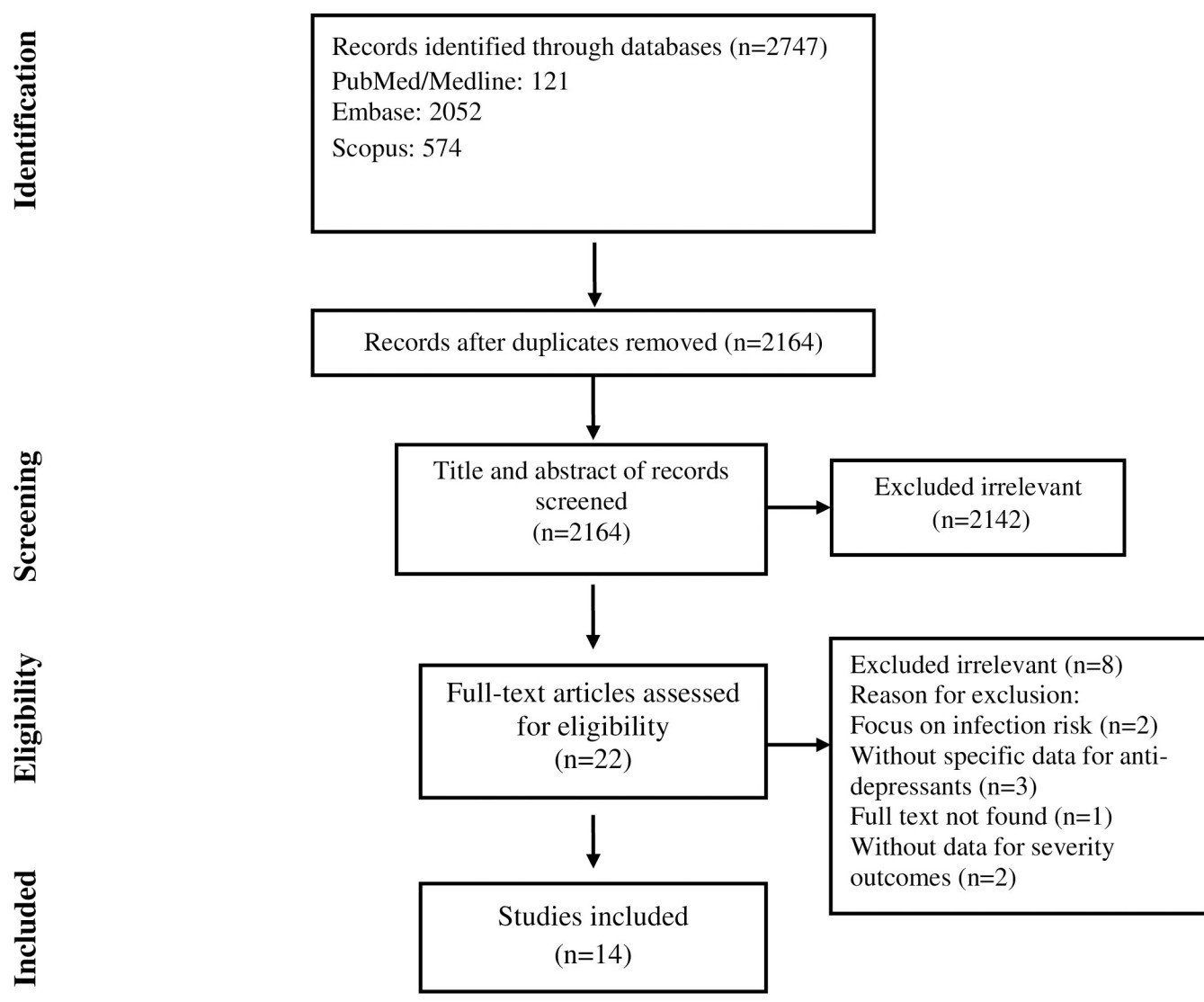

**Fig 1. Flow chart of study selection for inclusion in the systematic review and meta-analysis.**

## Results

The selection process of articles is shown in Fig 1. In this process Fourteen articles were included and classified into the followings: six cohort studies [8, 17–21], three case-control studies [22–24], and five clinical trials [25–29]. There were 1170 cases and 1180 controls in clinical trials, 15950 cases and 96638 controls in the cohort studies, and 97,844 cases and 80518 controls in the case-control studies, with a total population of 293300 in the whole studies (114964 cases and 178336 controls). Five studies were conducted in the USA and others in France, Turkey, Brazil, Hungary, Israel, Croatia, Scotland, Italy, and Korea. The duration of studies, detection test of COVID-19, and other study characteristics are shown in Table 1.

### Quality of the included studies

The checklists for observational studies [14] showed that the included observational studies had a low risk of bias except Bora et al. [17] and McKeigue et al. [23] studies (Tables 2 & 3). In contrast,

**Table 1. Study characteristics.**

| First author | Study design | Publication year | country | Study duration | detection test of COVID-19 |
|---|---|---|---|---|---|
| Lenze et al. [26] | Randomized Clinical Trial | 2020 | USA | April 10, 2020, to August 5, 2020 | PCR |
| Lenze et al. [28] | Randomized Clinical Trial | 2021 | USA | December 22, 2020, to September 28, 2021 | per lab or physician report based on PCR |
| Reis et al. [27] | Randomized Clinical Trial | 2021 | Brazil | June 2, 2020 to August 5, 2021 | RT-PCR |
| Calusic et al. [25] | cohort trial | 2021 | Croatia | April and May 2021, | PCR |
| Seo et al. [29] | Randomized Controlled Trial | 2022 | Korea | January 15, 2021, to February 19, 2021 | RT-PCR |
| Israel et al. [22] | case-control | 2021 | Israel | November 30, 2020 to December 31, 2020. | RT-PCR |
| McKeigue et al. [23] | case-control | 2021 | Scotland | June 6, 2020 to June 14, 2020 | N/R |
| Nemeth et al. [24] | case-control | 2021 | Hungary | March 17, 2021 to April 22, 2021 | antigen or polymerase chain reaction test |
| Rauchman et al. [20] | retrospective cohort | 2021 | USA | March 2020 to March 2021 | N/R |
| Hoertel et al. [8] | retrospective cohort | 2021 | France | January 24, 2020 to April 1, 2020 | RT-PCR |
| FEI et al. [18] | Prospective Cohort | 2021 | Italy | N/R | N/R |
| Oskotsky et al. [19] | retrospective cohort | 2020 | USA | January to September 2020 | laboratory test for SARS-CoV-2 (nucleic acid amplification tests and immunoassays) and/or by ICD-10 (for COVID-19 confirmed by laboratory testing) |
| Seftel et al. [21] | Prospective Cohort | 2021 | USA | November–December 2020 | PCR |
| Bora et al. [17] | retrospective cohort | 2021 | Turkey | October 1, 2020 to January 1, 2021 | the ICD-10 classification confirmed by the test results |

Abbreviations

PCR: Polymerase chain reaction/ RT-PCR: Reverse transcription polymerase chain reaction/ ICD 10: International Statistical Classification of Diseases and Related Health Problems, Tenth Revision/ N/R: Not reported

the checklist for experimental studies [14] showed that the included clinical trials had a low risk of bias except Calusic et al. [25] which had a high risk of bias for randomization, group concealment and participants, the treatment delivery and outcome assessors blinding (Table 4).

## Patient characteristics

According to eleven studies [8, 17–19, 22–26, 28, 29], the mean age of total patients was 58 years old, with 40 and 56.5 years old in case and control groups in nine studies, respectively [8, 18, 19, 21, 22, 24–26, 29]. 49.6 percent of patients were female, according to thirteen studies [8, 17–22, 24–29]. According to ten studies, this number was 52% in case groups and 49.3% in control groups [18–22, 24–27, 29]. Cases and controls were matched with age, sex, different kinds of comorbidities, smoking status, BMI category, etc. The complete definition of case and control groups and severity outcomes are shown in Table 5.

## Interventions and exposures characteristics

Nine studies [19, 21, 22, 24–29] only used or surveyed SSRIs as an anti-depressant in case groups (fluvoxamine in seven studies [19, 21, 25–29], fluoxetine in two studies [19, 24] and

**Table 2. Quality assessment of cohort studies.**

| author | 1 | 2 | 3 | 4 | 5 | 6 | 7 | 8 | 9 | 10 | 11 |
|---|---|---|---|---|---|---|---|---|---|---|---|
| Rauchman et al. [20] | yes | yes | yes | no | no | yes | yes | yes | yes | yes | yes |
| Hoertel et al. [8] | yes | yes | yes | yes | yes | yes | yes | yes | yes | no | yes |
| FEI et al. [18] | yes | yes | yes | no | no | yes | yes | yes | yes | yes | yes |
| Oskotsky et al. [19] | yes | yes | yes | no | yes | unclear | yes | yes | yes | yes | yes |
| Seftel et al. [21] | yes | yes | yes | no | yes | yes | yes | no | yes | yes | yes |
| Bora et al. [17] | yes | yes | yes | no | no | yes | yes | unclear | unclear | unclear | yes |

1. Were the two groups similar and recruited from the same population?

2. Were the exposures measured similarly to assign people to both exposed and unexposed groups?

3. Was the exposure measured in a valid and reliable way?

4. Were confounding factors identified?

5. Were strategies to deal with confounding factors stated?

6. Were the groups/participants free of the outcome at the start of the study?

7. Were the outcomes measured in a valid and reliable way?

8. Was the follow-up time reported and sufficient to be long enough for outcomes to occur?

9. Was follow-up complete, and, if not, were the reasons to loss to follow-up described and explored?

10. Were strategies to address incomplete follow-up utilized?

11. Was appropriate statistical analysis used

Escitalopram in one study [22]). In the remaining five studies [8, 17, 18, 20, 23] different types of SSRIs, SNRIs, and TCAs were used or surveyed. Anti-depressant type, dosage, frequency, and follow-up time are shown in Table 6.

## Effect of anti-depressants on severity outcomes of COVID-19

After adjustment, seven studies [8, 17, 19, 21, 22, 24, 26] showed a significant association between antidepressant use and reducing severity outcomes of COVID-19. On the other hand, five studies [18, 20, 23, 25, 27] presented no significant effects of anti-depressants in reducing disease severity or that anti-depressant are associated with severe COVID-19 in hospitalized

**Table 3. Quality assessment of case-control studies.**

| author | 1 | 2 | 3 | 4 | 5 | 6 | 7 | 8 | 9 | 10 |
|---|---|---|---|---|---|---|---|---|---|---|
| Israel et al. [22] (Cohort1) | yes | yes | yes | yes | yes | yes | yes | yes | yes | yes |
| Israel et al. [22] (Cohort2) | yes | yes | yes | yes | yes | yes | yes | yes | yes | yes |
| McKeigue et al. [23] | yes | yes | no | no | no | yes | no | no | yes | yes |
| Nemeth et al. [24] | yes | yes | yes | yes | yes | no | no | yes | yes | yes |

1. Were the groups comparable other than the presence of disease in cases or the absence of disease in controls?

2. Were cases and controls matched appropriately?

3. Were the same criteria used for identification of cases and controls?

4. Was exposure measured in a standard, valid and reliable way?

5. Was exposure measured in the same way for cases and controls?

6. Were confounding factors identified?

7. Were strategies to deal with confounding factors stated?

8. Were outcomes assessed in a standard, valid and reliable way for cases and controls?

9. Was the exposure period of interest long enough to be meaningful?

10. Was appropriate statistical analysis used?

**Table 4. Quality assessment of the clinical trials.**

| author | 1 | 2 | 3 | 4 | 5 | 6 | 7 | 8 | 9 | 10 | 11 | 12 | 13 |
|---|---|---|---|---|---|---|---|---|---|---|---|---|---|
| Lenze et al. [26] | yes | yes | yes | Yes | yes | yes | yes | yes | yes | yes | yes | yes | yes |
| Lenze et al. [28] | yes | yes | yes | Yes | yes | yes | yes | yes | yes | yes | yes | yes | yes |
| Calusic et al. [25] | unclear | unclear | yes | Unclear | unclear | unclear | yes | yes | yes | yes | yes | yes | yes |
| Reis et al. [27] | yes | yes | yes | yes | yes | yes | yes | yes | yes | yes | yes | yes | yes |
| Seo et al. [29] | yes | yes | yes | Yes | no | no | yes | yes | yes | yes | yes | yes | yes |

1. Was true randomization used for assignment of participants to treatment groups?

2. Was allocation to treatment groups concealed?

3. Were treatment groups similar at baseline?

4. Were participants blind to treatment assignment?

5. Were those delivering treatment blind to treatment assignment?

6. Were outcome assessors blind to treatment assignment?

7. Were treatment groups treated identically other than the intervention of interest?

8. Was follow-up complete, and, if not, were differences between groups in terms of their follow-up adequately described and analyzed?

9. Were participants analyzed in the groups to which they were randomized?

10. Were outcomes measured in the same way for treatment groups?

11. Were outcomes measured in a reliable way?

12. Was appropriate statistical analysis used?

13. Was the trial design appropriate and were any deviations from the standard randomized controlled trial design accounted for in the conduct and analysis of the trial?

patients. In the study which was conducted by Bora et al. [17], anti-depressants were associated with reducing mortality, regardless of the anti-depressant type. Hoertel et al. [8] suggested that antidepressant use (HR:0.56; 95% CI: 0.43–0.73) is significantly and substantially associated with reduced risk of intubation or death, independently of patient characteristics, clinical, biological markers of disease severity, and other psychotropic medications. They found that SSRI (HR:0.51; 95% CI:0.316–0.72) and non-SSRI (HR:0.65; 95% CI, 0.45–0.93) antidepressants, and specifically the SSRIs escitalopram, Fluoxetine, and Paroxetine, the SNRI venlafaxine, and the α2-antagonist antidepressants mirtazapine are significantly associated with reduced risk of intubation or death. Escitalopram is also effective in reducing the severity, according to Israel et al. [22] study. Fluoxetine use was associated with a significant (70%) decrease in mortality (OR:0.33; 95% CI:0.16–0.68) and threefold survival in the fluoxetine group, according to Nemeth et al. [24] study. Also, another study by Oskotsky et al. [19] showed similar results that Fluoxetine could reduce mortality risk (RR:0.72; 95% CI:0.54–0.97). On the other hand, Rauchman et al. [20] mentioned that prior use of SSRIs or SNRIs did not reduce mortality (OR:0.96; 95% CI:0.79–1.16). McKeigue et al. [23] proposed that TCAs (RR:1.1; 95% CI: 0.94–1.27) and SSRIs (RR:1.18; 95% CI:1.03–1.36), like other anti-depressants (RR: 1.76; 95% CI:1.5–2.07) increase the risk of severe COVID-19 because of their anticholinergic effect that is likely to increase risk of pneumonia. However, Fei et al. [18] represented that mortality rate, respiratory failure related to pneumonia, and renal failure as comorbidity within the anti-depressant treated subgroup is not lower than in the other patient of the sample but suggested that the anti-depressant treated patient subgroup show high mean age and large size of medical comorbidity. Although it is shown in this study that ARDS is significantly lower for the anti-depressant treated subgroup, so mild significant lower employment of protease inhibitors and endotracheal intubation is needed in this subgroup because of a lower level of IL-6 in anti-depressant treated patient that resulted from inhibition function of acid sphingomyelinase induced by antidepressants.

**Table 5. Patient characteristics.**

| First author | Age | | Gender (F/M %) | | Matching and analysis adjustment | Case definition | Control definition | Severity indices |
|---|---|---|---|---|---|---|---|---|
| | case | control | case | Control | | | | |
| Lenze et al. [26] | Mean:46 | | 71.7/28.3 | | 1:1 | Adults with SARS CoV-2 infection who received Fluvoxamine and were symptomatic within 7 days of the first dose of study medication | Adults with SARS CoV-2 infection who received placebo capsules and were symptomatic within 7 days of the first dose of study medication | oxygen saturation <92% plus supplemental oxygen needed and hospitalization related to dyspnea or hypoxia and/or ventilator support needed for ≥3 days |
| | 46 | 45 | 70/30 | 74/26 | Age, sex/Not-adjusted | | | |
| Lenze et al. [28] | Mean: 47 | | 62/38 | | 1:1 | unvaccinated positive test patients with ≤6 days symptoms and age≥30 that allocated for fluvoxamine | unvaccinated positive test patients with ≤6 days symptoms and age≥30 that allocated for placebo | Hospitalization or new hypoxemia within 15 days |
| | | | | | NR/Not-adjusted | | | |
| Reis et al. [27] | Median:50 | | 57.5/42.5 | | 1:1 | patients were allocated to fluvoxamine | patients were allocated to placebo | Hospitalized for COVID-19, need to mechanical ventilation and/or mortality |
| | | | 55/45 | 60/40 | BMI, Age, co-morbidities/Adjusted for comorbidities | | | |
| Calusic et al. [25] | Mean:65.7 | | 33.3/66.7 | | 1:1 | patients over the age of 18 with positive SARS-CoV-2 PCR test and acute clinical condition consistent with COVID-19 requiring ICU admission who received fluvoxamine + standard therapy | patients over the age of 18 with positive SARS-CoV-2 PCR test and acute clinical condition consistent with COVID-19 requiring ICU admission who received standard therapy | Days of hospital stay, Days of ICU stay, Days on ventilator support and/or mortality |
| | 65.6 | 65.9 | 33/67 | 33/67 | Age, sex, vaccination status, disease severity, comorbidities/Adjusted for comorbidities | | | |
| Seo et al. [29] | Mean: 53.5 | | 40.4/59.6 | | 1:1 | Patients who had symptoms with onset less than 7 days after randomization and had positive RT-PCR results and allocated to fluvoxamine | Patients who had symptoms with onset less than 7 days after randomization and had positive RT-PCR results and allocated to placebo | Decrease in O2 saturation (SpO2 <94%) with or without need to oxygen therapy, WHO Clinical Progression Scale ≥4, days to clinical deterioration |
| | 54 | 51.5 | 30.7/69.3 | 50/50 | Age/Not-adjusted | | | |
| Israel et al. [22] Group 1 | Mean: 64.7 | | 49.9/50.1 | | 1:5 | Hospitalized COVID-19 patients | patients were chosen among the general population | overall risk for hospitalization due to COVID-19 |
| | 64.6 | 64.8 | 50/50 | 50/50 | Age, sex, comorbidities, BMI, smoking status/Not-adjusted | | | |
| Israel et al. [22] Group 2 | Mean: 65.7 | | 48.6/51.4 | | 1:2 | Hospitalized COVID-19 patients | patients who had a positive test for SARS-CoV-2 but had not been hospitalized | Risk for COVID-19 hospitalization in patients who had a proven infection with the virus. |
| | 65.7 | 65.7 | 49/51 | 49/51 | Age, sex, comorbidities, BMI, smoking status/Not-adjusted | | | |
| McKeigue et al. [23] | Mean:61.2 | | N/R | | 1:10 | A positive nucleic acid test followed by entry to critical care or death within 28 days or a death certificate with COVID-19 as underlying cause. | Controls were alive and had not yet tested positive on the date that the case first tested positive. | entry to critical care and/or mortality |
| | | | | | Sex/Adjusted for comorbidities | | | |
| Nemeth et al. [24] | Mean: 66 | | 54.6/45.4 | | 1:1.5 | adult patients hospitalized with moderate or severe COVID-19 pneumonia who received Fluoxetine as an adjuvant medication in combination with antiviral drugs | adult patients hospitalized with moderate or severe COVID-19 pneumonia who received antiviral drugs | mortality between hospital days 2 and 28 |
| | | | | | N/R | | | |
| | 65 | 67 | 53/47 | 44/56 | Not-adjusted | | | |

(*Continued*)

**Table 5.** (Continued)

| First author | Age | | Gender | | Matching and analysis adjustment | Case definition | Control definition | Severity indices |
|---|---|---|---|---|---|---|---|---|
| | | | (F/M %) | | | | | |
| | case | control | case | Control | | | | |
| **Rauchman et al. [20]** | > = 18 | | 45.9/54.1 | | 1:10 | adult patients 18 and over admitted with a diagnosis of COVID-19 and on an antidepressant drug during admission | adult patients 18 and over admitted with a diagnosis of COVID-19 and not on an antidepressant drug during admission | mortality |
| | | | | | Age, sex, primary race/ | | | |
| | | | 61/39 | 44/56 | Not-adjusted | | | |
| **Hoertel et al. [8]** | Mean:57.6 | | 49.2/50.8 | | 1:14.5 | Patients who receiving any antidepressant during the first 48 h of hospital admission and before the end of the index hospitalization or intubation or death | Patients not receiving any antidepressant during the first 48 h of hospital admission and before the end of the index hospitalization or intubation or death | intubation or death |
| | 73.7 | 56.8 | | | Age, sex, obesity, smoking status/ Adjusted for comorbidities, medical indication and baseline severity | | | |
| **FEI et al. [18]** | Mean:70 | | 40.3/59.7 | | 1:11 | patient treated with an antidepressant before admission until discharge from hospital | patient not treated with antidepressant | noninvasive ventilation (NIV), intubation, ICU admission, mortality |
| | 80.1 | 69.1 | 56/44 | 39/61 | N/R | | | |
| | | | | | Not-adjusted | | | |
| **Oskotsky et al. [19]** | Mean:52 | | 50.1/49.9 | | 1:21 | Patients with COVID-19 and a medication order for an SSRI at least once within a period of 10 days before and 7 days after their first recorded COVID-19 diagnosis | patients with COVID-19 and no SSRI orders | mortality |
| | | | | | Age, sex, race & ethnicity, comorbidities | | | |
| | 63.1 | 51.6 | 59/41 | 50/50 | Adjusted for comorbidities and medical indication | | | |
| **Seftel et al. [21]** | Median:42 | | 24.8/75.2 | | 1.5:1 | patients treated with fluvoxamine for 14 days | patients not treated with fluvoxamine | Hospitalization, CU care and /or mortality |
| | Mean:44 | Mean:43 | 23/77 | 27/73 | Demographic features/Not-adjusted | | | |
| **Bora et al. [17]** | Mean:47.6 | | 46.6/53.4 | | 1:9 | patients treated by antidepressants with no limit to the types of these medications | patients not treated by antidepressants | mortality |
| | | | | | N/R | | | |
| | | | | | Not-adjusted | | | |

Abbreviations

BMI: Body mass index/ ICU: Intensive care unit/ PCR: Polymerase chain reaction/ N/R: Not reported/ HTN: Hypertention/ CAD: Coronary artery disease/ COPD: Chronic obstructive pulmonary disease/ CVD: Cerebrovascular disease/ CKD: Chronic kidney disease/ IHD: Ischemic heart disease/

## The specific effect of fluvoxamine on severity outcomes of COVID-19

Six studies have discussed the relation between using fluvoxamine and COVID-19 severity. One cohort study [21] and five clinical trials [25–29]. Lenze et al. [26] represented that patients treated with fluvoxamine, compared with placebo, had a lower likelihood of clinical deterioration over 15 days (absolute difference:8.7% 95% CI:1.8%-16.4%). Also, severe adverse events in the fluvoxamine group were less than in the placebo group (1.3% vs 6.9%). They suggested that this effect is because of the influence of fluvoxamine on the S1R-IRE1 pathway and anti-inflammatory (Cytokine reduction) actions resulting from S1R activation. Also, according to Seftel et al. [21] study, the incidence of subsequent hospitalization was lower in the fluvoxamine group (0% vs. 12.5%) and elevated respiratory rates were improved faster by day 7 in this group, so fluvoxamine seems to be promising as it use in early treatment for COVID-19 to prevent clinical deterioration requiring hospitalization and to prevent possible long haul symptoms persisting beyond two weeks. On the other hand, Reis et al. [27] suggested that There were no significant differences between fluvoxamine and placebo in secondary outcomes like

**Table 6. Interventions and exposures characteristics.**

| First author | Anti-depressant type | Dosage | Frequency | Follow-up time |
|---|---|---|---|---|
| Lenze et al. [26] | Fluvoxamine | 50 mg (in the evening immediately after the baseline assessment and confirmation of eligibility) | — | 30 days |
| | | 100 mg (for 2 days) | twice daily | |
| | | 100 mg (through day 15 then stopped) | 3 times daily | |
| Lenze et al. [28] | Fluvoxamine | 100 mg (15 days) | twice daily | 90 days |
| Reis et al. [27] | Fluvoxamine | 100mg (for 10 days) | twice daily | 28days |
| Calusic et al. [25] | Fluvoxamine | 100 mg (for 15 days) | 3 times daily | 22 days |
| | | 50 mg (After Day15) | twice daily | |
| Seo et al. [29] | Fluvoxamine | 100 mg (for 10 days) | twice daily | 30 days |
| Israel et al. [22] | Escitalopram | N/R | N/R | 35 days |
| McKeigue et al. [23] | TCAs & SSRIs & other anti-depressants | N/R | N/R | N/R |
| Nemeth et al. [24] | Fluoxetine | 20 mg | once daily | N/R |
| Rauchman et al. [20] | Citalopram, Desvenlafaxine, Duloxetine, Escitalopram, Fluoxetine, Paroxetine, Sertraline, Venlafaxine | N/R | N/R | N/R |
| Hoertel et al. [8] | Citalopram | 20 mg | once daily | 6 days |
| | Escitalopram | 10 mg | once daily | 13 days |
| | Flouxetine | 20 mg | once daily | 13.5 days |
| | Paroxetine | 20 mg | once daily | 8 days |
| | Sertraline | 50 mg | once daily | 8.5 days |
| | fluvoxamine | NR | NR | NR |
| | Vortioxetine | 10 mg | once daily | NR |
| | Venlafaxine | 75 mg | once daily | 28 days |
| | Duloxetine | 60 mg | once daily | 8 days |
| | Amitriptyline | 15 mg | once daily | 7 days |
| | Clomipramine | 12.5 mg | once daily | 10 days |
| | Mianserin | 30 mg | once daily | 10 days |
| | Mirtazapine | 15 mg | once daily | 11 days |
| FEI et al. [18] | Sertraline | N/R | N/R | 3 months |
| | Escitalopram | | | |
| | Citalopram | | | |
| | Paroxetine | | | |
| | Venlafaxine | | | |
| | Duloxetine | | | |
| | Escitalopram + Venlafaxine | | | |
| Oskotsky et al. [19] | Fluoxetine | 30.2mg/d | N/R | 8 months |
| | Flouxetine or Fluvoxamine | 29mg/d | | |
| | SSRI other than Flouxetine or Fluvoxamine | 30.4mg/d | | |
| Seftel et al. [21] | Fluvoxamine | 50- to 100-mg loading dose | — | 14 days |
| | | 50 mg | twice daily | |
| Bora et al. [17] | Duloxetine, Escitalopram, Fluoxetine, Fluvoxamine, Mirtazapine, Paroxetine, Sertraline, Venlafaxine, tricyclic antidepressants | N/R | N/R | N/R |

Abbreviations

SSRI: Selective serotonin reuptake inhibitors/ TCA: Tricyclic antidepressant

viral clearance at day 7 (RR:0·67; 95% CI:0·42–1·06), hospitalizations due to COVID-19 (RR:0·77; 95% CI:0·55–1·05), time to hospitalization (RR:0·79; 95% CI:0·58–1·06), mechanical ventilation (RR:0·77; CI:0·45–1·30), time on mechanical ventilation (RR:1·03; 95% CI: 0·64–1·67), death (RR:0·69; 95% CI:0·36–1·27) and time to death (RR:0·80; 95%CI:0·43–1·51). There were significant differences between the two groups in the primary outcome (hospitalization defined as either retention in a COVID-19 emergency setting or transfer to the tertiary hospital due to COVID-19, RR:0·69; 95%CI: 0·53–0·90) and according to a separate letter [30], the primary authors of this article clarified that if they apply other composite endpoints like hospitalization or emergency care >24 h (RR:0·74; 95% CI:0·56–0.98) or the outcome for FDA criterion for severe COVID-19 (RR:0·67; 95% CI:0·52–0.86), their effect size remains significant. Also, their per-protocol analysis indicates a significant reduction in mortality (RR: 0·09; 95% CI: 0·01–0.47). However, according to Calusic et al. [25] study, no statistically significant differences between groups were observed regarding the number of days on ventilator support, duration of ICU or total hospital stay. Overall mortality was lower in the fluvoxamine group (HR: 0.58; 95% CI: 0.36–0.94). Although only mortality in women was significantly reduced in the fluvoxamine group (HR: 0.40; 95% CI: 0.16–0.99) and there were no significant differences in mortality of men (HR: 0.69; 95% CI: 0.39–1.24). Also, Seo et al. [29] mentioned that none of their outcomes (Decrease in O2 saturation (SpO2 <94%) with or without need for oxygen therapy, WHO Clinical Progression Scale ≥4 and days to clinical deterioration) differed significantly between the treatment groups and Lenze et al. [28] proposed that there were not any significant difference between two groups in Hospitalization or new hypoxemia within 15 days (RR:0.93; 95% CI: 0.42–2.06).

## Statistical focus on fluvoxamine

The meta-analysis of clinical trials showed that fluvoxamine could significantly decrease the severity outcomes of COVID-19 (RR: 0.763; 95% CI: 0.602–0.966, I2: 0.0) (Fig 2). There was no evidence of publication bias (p > 0.05; Begg: 0.80, Egger: 0.49) (Fig 3).

## Discussion

In the present systematic review study, we included cohort, case-control, and clinical trials that contained any treatment outcome with antidepressant drugs. We did not aim for the psychiatric findings of the drugs. In fact we compared the effect of antidepressant drugs on severity and mortality in Covid-19 patients. In the present study, most experiments revealed that the

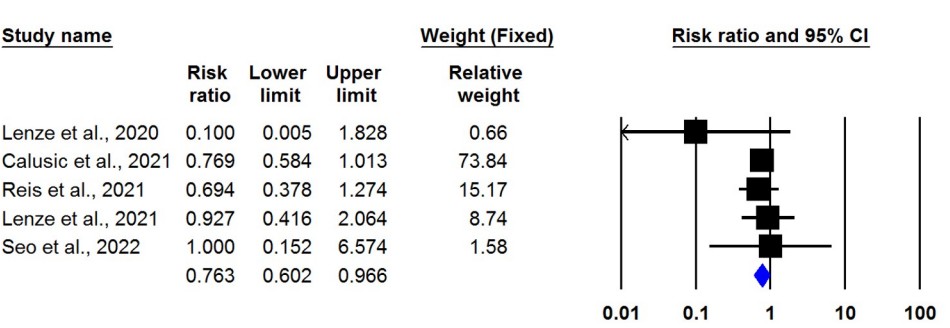

**Fig 2. Pooled RR for clinical trials.**

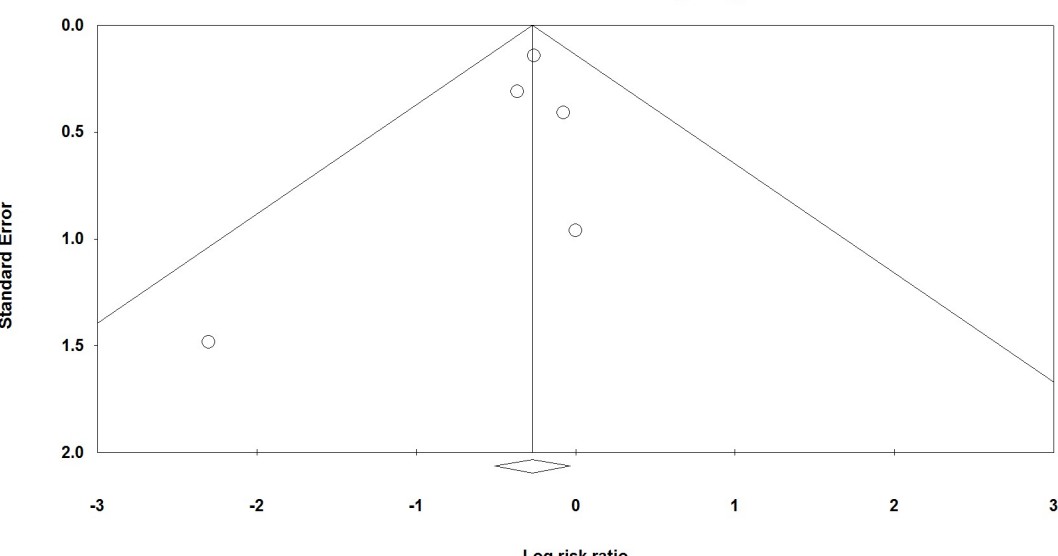

**Fig 3. The funnel plot of analysis.**

antidepressant medication subgroups generally reduce the risk of ARDS and consequent death in hospitalized Covid-19 patients compared to control groups. However, conversely, some studies reported no difference between case and control groups, or even one study proposed that antidepressants such as SSRIs and TCAs increase the risk of severe COVID-19 due to their anticholinergic effect can attributable to cause pneumonia. Although, as mentioned in Table 5 and the quality assessment section, this study was adjusted only for comorbidities but baseline severity and medical indication of antidepressant usage were not considered for adjustment, so the results of this high risk of bias article is unreliable.

Our meta-analysis on clinical trials also revealed that fluvoxamine could significantly decrease the severity outcomes of COVID-19. Seftel et al. [21] proposed that fluvoxamine can be an effective therapeutic medication to prevent clinical deterioration, hospitalization, intubation, and death for infected patients. Next, we will review and evaluate the pieces of evidence to determine which can complete our experimentally achieved puzzle on the molecular aspect through the antidepressant medication's mechanisms of action.

## Depression and imuunemediated disorders

It is proven that psychiatric disorders like stress or depression, through the transmission of nerve signals to the hypothalamus, cause the secretion of CRH, ACTH, and glucocorticoids. Glucocorticoids are an immunosuppressive agent on the immune cells by inhibiting cytokine production needed for inflammation development. These cytokines contain tumor necrosis factor (TNF)-α, IL-1, IL-2, IL-6, and IFN-γ, leading to reduced T lymphocyte activity [31]. So it is predictable to have fewer circulating CD4+ T-cells and a reduction in natural killer cell and cytotoxic lymphocyte proliferation responses in elderly patients with clinical depression and stressful states. So abnormal regulation of the hypothalamus-pituitary-adrenal axis, sympathetic–adrenal–medullary axis, and hypothalamic-pituitary-ovarian axis in patients with clinical depression can be the other aspects of this puzzle [6].

On the other hand, severe cases of COVID-19 infection are associated with increased anti-inflammatory cytokines, including IL-6, IL-8, IL-10, and also the proliferation of CD8

+ cytotoxic and CD4+ T-cells, which are involved in neutralizing the virus particles and inflammation of the lung [32]. Thus depression inhibits this natural event by suppressing the production of cytokines. In Covid-19 patients with clinically diagnosed depression, anti-depression treatment could improve immune responses by neutralizing this comorbidity [33]. So using antidepressants of any type can improve mental health, reduce hypothalamic-pituitary-adrenal malfunction and reduce immunosuppression. It should be considered that in RCT studies, the duration of antidepressant consumption is a short time to treat depression, but in retrospective studies, better mental health after antidepressant treatment can be supposed that less immunocompromised immune system and better outcomes is predictable.

Still, there are several other mechanisms through which antidepressants might lower the risk of severe outcomes.

### Direct effect on serotonin transporters

SSRI and SNRIs are inhibitors of serotonin receptors. SSRI and SNRIs have shown anti-inflammatory effects in both preclinical and clinical studies [11, 34, 35]. This is attributable to the serotonin transporter inhibition activity [36]. Serotonin is involved in anti-inflammatory cytokine production [37]. For example, it can increase the release of IL-10, as a potent cytokine with reputable anti-inflammatory aspects [38], and moderate TNF-$\alpha$ and IL-6 [38], Which are both responsible for the inflammatory response in Covid-19 infection. Serotonin also inhibits the production of inflammatory cytokines such as interferon-gamma (IFN-$\gamma$) by human blood leucocytes [39, 40]. High serotonin transporter expression in the lungs [41] insist in its possible function in lung inflammation. Serotonin reduces IL-12 and TNF-$\alpha$ release in human alveolar macrophages but increases IL-10 production via 5-HT2 receptors [42]. Therefore, SSRIs and SNRIs may influence directly on cytokine storm suppressing in lungs. Considering that mentioned anti-inflammatory effects were not observed in other antidepressants which were able to block serotonin transporters, the role of serotonin transporter inhibition in beneficial effects of SSRIs for COVID-19 patients seems doubtful. However, it might be a probable factor in the favorable effects of SSRIs such as fluoxetine and fluvoxamine [36].

### Platelet aggregation

As it has known, SSRI medications inhibit serotonin reuptake in the brain and the peripheral tissues, including the platelet cells. Platelets transport serotonin, which is secreted during the activation time to contract injured blood vessels to minimize blood loss. [43] on the other hand during the thrombosis process, platelets release serotonin which causes more platelet aggregation. In a study held by Neha Rastogi-Chawla it was demonstrated that whole blood platelet aggregation was decreased in 17.4% of patients with SSRI usage. (p = 0.035) [44]. Also, Donna Jo McCloskey et al. demonstrated that SSRIs interfere with platelet aggregation. [45] even antidepressants like Citalopram inhibits platelet function independently of the SERT-mediated mechanism. So there is a consensus that SSRI medications interfere with platelet aggregation [46] hyper coagulopathy in Covid-19 patients is common and accounts for many patients' death. So this anti-platelet function of SSRIs can benefit in preventing complications of the hypercoagulopathy and pulmonary thromboembolism, but this is a weak hypothesis because anit platelet function of SSRIs appears in long-term use of them. Of course, this issue can be attributable to the patients in retrospective studies with a history of longterm SSRI consumption.

### Acid sphingomyelinase (ASM)

Binding the S1 antigen of the SARS-CoV-2 virus spike glycoprotein to its receptor is the first step in entering the virus two the cell. This receptor, known as angiotensin-converting enzyme

2 (ACE2) leads to the cleavage of the viral spike protein by the activity of transmembrane serine protease 2 (TMPRSS2) or cathepsin L and viral entry [46, 47]. On the other hand, the presence of active acid sphingomyelinase on the cell surface leads to the hydrolysis of sphingomyelin and its conversion to ceramide in the outer cell membrane. [48, 49] hydrophobic nature of ceramide molecules and consequent formation of small ceramide-enriched membrane domains causes to alternate biophysical features of the plasma membrane. These molecules make the tightly dense, gel-like ceramide-enriched membrane domains which play an important role in the entrance of the virus to the target cells [50].

Some potent studies revealed that plasma ceramide levels could strongly correlate with clinical severity among patients with Covid-19. Mehran M. Khodadoust et al., in the study of plasma samples from 52 patients, demonstrated that infected patients have 400 times more ceramide molecules level than healthy people [51]. This matter is important and can even predict disease severity by analyzing the serum sphingolipids levels. In an investigation held by Enrica Torretta et al., the sphingolipid profile of twenty-four healthy and fifty-nine COVID-19 patients was analyzed and the correlation between the level of ceramide with the disease's severity demonstrated [52]. This increase in ceramide level can be attributable to increased activity of acid and neutral sphingomyelinases(ASM, NSM) or a reduction in neutral ceramidase (NC) activity [53].

Some Preclinical studies in both in vitro and ex vivo suggest that inhibition of the ASM/ceramide system can be potentially beneficial in reducing the severity of the disease. Moreover, it may explain both potential antiviral and anti-inflammatory effects of certain antidepressants known as FIASMA (Functional Inhibitor of Acid SphingoMyelinAse) in Covid-19 [54, 55]. FIASMA medications are various range of medications, including antidepressants, such as Fluoxetine, Fluvoxamine, Sertraline, Paroxetine in the group of SSRI or SNRI and also Amitriptyline, Nortriptyline, Protriptyline Desipramine, Imipramine Trimipramine, Maprotiline as non-SSRI antidepressants that directly inhibit acid sphingomyelinase enzyme [11, 56–58] beyond the mentioned medications, FIASMA also contains a variety of non-anti-depressant medications which the most known drugs include Amlodipine, Biperiden, Carvedilol, Chlorpromazine, Clomipramine, Doxepin, Fluphenazine, Hydroxyzine, Loperamide, Loratadine, Perphenazine, Promethazine, Trifluoperazin, Triflupromazine, and Tamoxifen [58].

Alexander Carpinteiro et al., in an in-vitro study, demonstrated that Amitriptyline, even in low doses, provides long-lasting and very efficient protection against the cultured cells of isolated human nasal epithelial cells. Also, they proved that Amitriptyline, Imipramine, Fluoxetine, Sertraline, Escitalopram, or Maprotiline or genetic downregulation of the enzyme has the same effect and prevents infection with a pp-VSV-SARS-CoV-2 spike [59].

Ambroxol(trans-4-[(2, 4-dibromanilin-6-yl)-methyamino]-cyclohexanol) is a mucolytic medication that owns a FIASMA activity. This medication is applied by inhalation and is not an antidepressant, but the experiment held by Alexander Carpinteiro suggests that the drug might inhibit the acid sphingomyelinase and thereby limits the infection with SARS-CoV-2 [55]. So even only the ASM activity of a medication can be beneficial in preventing the viral replication without any antidepressant, anti-inflammatory, or serotoninogenet properties. Beyond them, for FIASMA antidepressants, anti-inflammatory and ASM activity can be considered together which anti-inflammatory properties are mentioned in the next.

The in vitro inhibition of ASM shows various potency among the SSRIs which Fluoxetine is the most potent, and  Paroxetine, Fluvoxamine  and other SSRIs are in the next rank [58]. This difference correlates with the outcomes of in vitro antiviral effect against SARS-CoV-2, which makes the present mechanism more probable [60, 61].

In this matter, we should consider a few ambiguous points that can complete the ASM mechanism puzzles. First Nortriptyline, Protriptyline, Desipramine, Imipramine,

Trimipramine, and Maprotiline are known as non-SSRI antidepressants; despite their FIASMA activities, there are not enough sufficient clinical evidence to verify their effect on Covid-19. We expected to hear some more! Of course, it needs more studies to prove this fact and a lack of information can't reject the hypothesis.

Second, as mentioned above, Chlorpromazine is a typical antipsychotic [62], but in an observational study of Chlorpromazine in Hospitalized Patients with COVID-19 on 14,340 patients, it was concluded that Chlorpromazine's daily prescription dose of 70.8 mg (SD 65.3) was not associated with reduced mortality [63]. Also, another study showed that Chlorpromazine, Fluphenazine, and Perphenazine are antipsychotic medications that stop clathrin-dependent endocytosis, cell-cell fusion, and replication of the virus and consequent decrease viral invasion as well as suppress entry into the host cells [64] but as mentioned in the study, chlorpromazine was not effective. In this case we should consider the dose-related effect, comorbidities of the patients, and some other cofounder conditions. For Chlorpromazine, it should be considered that the therapeutic serum level is between 30–300 ng/mL, but a study by M. Masson et al. proved that we have at least a 50–60% decrease in activity of ASM need about 25–50 µg/mL of the medication in serum. Of course, reaching this concentration is not possible in the human body because of its toxic effects like cardiac arrhythmia [58, 64], so it is essential to know that the effectiveness of a medication which was effective in an in-vitro study previously depends on the magnitude of the in an invitro experiment which it may not be reached in the clinical setting. Also it depends on the dose prescribed and the concentration in the site of action. For example, effective results of Fluoxetine in the treatment of Covid-19 have clinically relevant to pharmacogenomics concerns, and a minimum dose of 20 mg/day for at least 10-days inhibits SARS-CoV-2 infectivity due to efficient lung distribution at a 60-times higher concentration. However, this matter may not be achievable for other medications [65]. As we mentioned before, According to Nemeth et al. [24] study Fluoxetine use was associated with a about 70% decrease in mortality and threefold survival in the Fluoxetine group;. Also another study by Oskotsky et al. [19] showed a similar result that Fluoxetine could reduce risk of mortality.

In a study held the New York State on 165 patient it was demonstrated that in a psychiatric patients taking antidepressants particularly fluoxetine and trazodone, can reduce the infection risk and it can be protective [7]. Also Hoertel et al, in an study which was adjusted for sociodemographic characteristics, psychiatric and other medical comorbidity, and other medications concluded that role of the ASM/ceramide system framework in COVID-19 is important and continuation of FIASMA psychotropic medications in these patients is suggested.

Finally, determining whether the antiviral effect of FIASMA group medication relates to their direct activity needs more independent investigation with considering comorbidities.

## Sigma-1 receptor

Sigma-1 receptor (Sig-1R), which is known as ER chaperone protein, plays a regulatory role as an inter-organelle signaling modulator. This receptor usally resides on mitochondrion-associated ER membrane contact and modulates ER-mitochondrion signaling and ER-nucleus crosstalk [66]. This receptor regulates IP3 receptors and calcium signaling at the endoplasmic reticulum, movement of cytoskeletal adaptor proteins, nerve growth factor-induction, modulation of potassium channels as a regulatory subunit, alteration of psychostimulant-induced gene expression, and blockade of spreading depression and pain perception [66]. Some evidence shows that the sigma receptor can be effective on Covid-19 in clinical and molecular studies. In a study, Library of 290 compounds with FDA approval was studied for antiviral activity against both MERS-CoV and SARS-CoV viruses [67]. Twenty-seven compounds

displayed in vitro activity against MERS-CoV and SARS-CoV. Among the 27 active compounds, at least 19 of them could bind to Sig-1R with significant affinity (chloroquine, hydroxychloroquine, mefloquine, amodiaquine, tamoxifen, toremifene, terconazole, cycloheximide, benztropine, fluspirilene, thiothixene, fluphenazine, promethazine, astemizole, chlorphenoxamine, Chlorpromazine, thiethylperazine, triflupromazine and clomipramine) This common feature among most of the effective medications are notable. [68] of course, most of the medications had a Acid sphingomyelinase effect, too and it can support the previous ASM hypothesis, too.

When these antidepressants bind to the Sigma-1 receptor, their agonist effect leads to sigma-1 receptor chaperone activity, which protects against disease-related ER stress [69, 70]. Another study demonstrated that the binding of Fluoxetine to the sigma-1 receptor in the endoplasmic reticulum decreases cytokine activity and enhances survival in preclinical models of sepsis and inflammation [71]. Given the possible role of the chaperone activity by sigma-1 receptor agonist in SARS-CoV-2 replication, it might explain the effect of SSRIs (i.e., Fluvoxamine, Fluoxetine, Escitalopram) with sigma-1 receptor agonist mechanism, in reducing the severity of COVID-19 [8] but it remains indeterminate yet in the absence of any preclinical or clinical data specific to Covid-19 for supporting this effect. Noting the associations observed between non-S1R antidepressants (such as Paroxetine, mirtazapine or Venlafaxine) which have FIASMAs properties [58], and reduced risk of intubation or death as well as anti-inflammatory effects observed in a broad range of other antidepressants (not only Fluvoxamine and Fluoxetine), represent that S1R agonist effect is not a competent explanation [11].

## Other anti-inflammatory effects of anti-depressant

Toll-like receptor 4 activation medicates to activating the factor κ B. this factor is a vital element in the perception of exogenous oxidants. The TLR4-NF-kB interaction plays a crucial role in initiating infection-induced lung injury. SARS-CoV-2 infection in severe COVID-19 patients [72] Phosphatidylinositol 3-Kinase (PI3K) is a cell survival factor that inhibits premature apoptotic cell death during viral infections. In the downstream targets of the PI3K/AKT signaling pathway, mTOR and NF-κB are probable to lead the pathogenicity of SARS-CoV-2. For example, hyperactivity of NF-κB has been shown in SARS-CoV-2 infected cells on the other hand, inhibition of mTOR followed by the PI3K/AKT signaling pathway suppression has been associated with inhibitory effects on the life cycle of SARS-CoV-2 in different studies [73, 74]. This pathway leads to producing inflammatory cytokine.SSRI medication can inhibit the PI3K/AKT signaling pathway through GSK1 [75].

## Other probable mechanisms

It is demonstrated that fluvoxamine inhibits cytochrome P450 CYP1A2. This inhibition leads to melatonin as an anti-inflammatory, immunomodulatory, and antioxidant agent, which can cause an anti-inflammation in the lungs and reduce the consequent damages [76].

## Dealing with conflicting results

Among the mentioned articles with conflicting results like lack of significant effect for antidepressants, Fei et al. [18] and Calusic et al. [25] studies were limited by small sample size. Calusic et al. [25] and Reis et al. [27] studies could not entirely rule out selection bias. Fei et al. also stated that the control group showed higher mean age and large size of medical comorbidities, which each factor can be an independent cause of more mortality rate [18].

Another important clinical issue is to continue or discontinue SSRIs when a COVID-19 patient is hospitalized or admitted to an ICU. There is evidence showing adverse ICU

outcomes after discontinuation of SSRIs. It is probably due to increased agitation and the need for sedation among these patients that can result in respiratory depression [77, 78]. SSRI use may sometimes be contraindicated in ICU patients due to ECG changes and abnormal coagulation effects [79]. A systematic review study regarding the use of antidepressants in Critical Care suggested that there may be excess morbidity in critically ill SSRI or SNRI users, however, whether this is due to chronic effects, ongoing use, or drug withdrawal is unclear [78].

Reis et al. [27] study suggested the modulatory effect of fluvoxamine on systemic inflammation, including lower respiratory tract infections, and hospital admissions, were reported less frequently in patients in the fluvoxamine group compared to those in the placebo group.

Finally, the efficacy of fluvoxamine in COVID-19 patients could be rely on the initiating time of treatment which increased effectiveness could be achieved if treatment is initiated earlier during SARS-CoV-2 infection.

## Limitations

Most of the selected articles in this study were not limited only to the antidepressant medications and didn't studied each drug separately. They usually studied antidepressants in discriminate groups like SSRIs, SNRIs, and TCAs. We reviewed antidepressants but could only hold a meta-analysis for fluvoxamine. This medication was studied in five clinical trials but this investigation limited to a small population of 2350 patients. We also generally expressed our results about only outcomes of severity and couldn't arrange a subgroup analysis for each outcome or sex group. On the other hand people taking antidepressants like the cases in our reviewed studies, are more likely to have medical comorbidities than their counterparts. Therefore, comorbidities, indication of the treatment (i.e., depressive disorder, anxiety disorder, dementia, neurologic pain), or baseline severity of SARS-CoV-2 infection should be taken into account. But as it shown in Table 5, it should be considered that only few studies have adjusted the analysis by comorbidities,baseline severity of disease and indication of treatment. So conflicting results in our review can be attributable to this lack of adjustment.

## Suggestions

As there is strong evidence of the link between antidepressant use and improving outcomes of Covid-19, it seems legible to conduct more research on the subject, aiming to find more therapeutic options to treat Covid-19. Future studies should focus on antidepressants separately and be more specific about the outcome in different patient groups. We also need to arrange more clinical trials with larger populations to confirm the efficacy of a candidate certainly. SSRIs such as Fluoxetine and Fluvoxamine are supported by more substantial evidence and could be favorable options for future research programs. It also suggests that for prioritizing Fluoxetine and fluvoxamine; clinical trial studies will be held in the future.

In the molecular aspects other unpopular antidepressants' mechanisms are supposed to exist like non–S1R-inositol-requiring enzyme (IRE) pathways, Toll-like receptor 4 and factor κ B, PI3K/AKT/mTOR pathway and peroxisome proliferator-activated receptor γ which should be more investigated in future studies.

## Conclusion

Among the reviewed studies, Eight studies showed anti-depressant ability to reduce the severity of Covid-19, and five did not mention any significant effect. The Covid-19 reducing effect of the mentioned medications can be concluded from the decline in the risk of intubation and death, clinical outcomes, and biological markers demonstrating the disease's severity. Among the SSRI medications, fluvoxamine was the only antidepressant evaluated in clinical trials to

treat COVID, which has a less subsequent hospitalization rate, and less likelihood of 15-day-deterioration compared to the placebo. These results are attributable to the Acid sphingomyelinase activity, S1R-IRE1 pathway and the anti-inflammatory properties of antidepressants. On the other hand, the most studied antidepressant is Fluoxetine, with the most remarkable adequate size in observational studies, which is beneficial in this matter. Other SSRI medications, including Escitalopram and Paroxetine, and Venlafaxine as a SNRI, are also reasonably associated with a reduced risk of intubation or death. But they have not been studied as much as Fluvoxamine and Fluoxetine yet and these antidepressants should be prioritized in future clinical trials.

The most probable mechanisms of action for antidepressants included Sigma receptor 1 pathway, Acid sphingomyelinase activity inhibition and anti-inflammatory properties of antidepressants. So other mechanisms also can be assumped by non–S1R-inositol-requiring enzyme (IRE) pathways, such as nuclear factor κ B, Toll-like receptor 4, or peroxisome proliferator-activated receptor γ which should be more investigated in future studies.

## Supporting information

**S1 File. PRISMA 2020 checklist.**
(DOCX)

**S2 File. Data extraction form.**
(XLSX)

**S3 File. Search strategy of PubMed/Medline, Embase and Scopus.**
(DOCX)

## Acknowledgments

This study is related to the MPH project From the Department of Public Health, School of Public Health and Safety, Shahid Beheshti University of Medical Sciences, Tehran, Iran.

## Author Contributions

**Data curation:** Reza Bayati, Mohammad Rahmanian.

**Formal analysis:** Moein Zangiabadian.

**Investigation:** Reza Bayati, Mohammad Rahmanian, Amir Ghaffari Jolfayi.

**Methodology:** Moein Zangiabadian.

**Supervision:** Sakineh Rakhshanderou.

**Writing – original draft:** Hosein Nakhaee, Moein Zangiabadian.

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
