## [Decision Letter · Decision Letter 0]

10 Jun 2022

PONE-D-22-10307The effect of antidepressants on severity of COVID-19 in hospitalized patients:

A systematic review and meta-analysisPLOS ONE

Dear Dr. ZANGIABADIAN,

Thank you for submitting your manuscript to PLOS ONE. After careful consideration, we feel that it has merit but does not fully meet PLOS ONE’s publication criteria as it currently stands. Therefore, we invite you to submit a revised version of the manuscript that addresses the points raised during the review process.

Please revise.

We look forward to receiving your revised manuscript.

Kind regards,

Academic Editor

PLOS ONE

Journal Requirements:

2. Thank you for submitting the above manuscript to PLOS ONE. During our internal evaluation of the manuscript, we found significant text overlap between your submission and the following previously published works:

- https://covidebm.umn.edu/fluvoxamine

- https://www.nature.com/articles/s41380-021-01432-3

Please revise the manuscript to rephrase the duplicated text, cite your sources, and provide details as to how the current manuscript advances on previous work. Please note that further consideration is dependent on the submission of a manuscript that addresses these concerns about the overlap in text with published work.

5. Please upload a copy of Supporting Information Table S1 which you refer to in your text on page 3.

Reviewers' comments:

Reviewer's Responses to Questions

**Comments to the Author**

1. Is the manuscript technically sound, and do the data support the conclusions?

Reviewer #1: No

Reviewer #2: Yes

2. Has the statistical analysis been performed appropriately and rigorously? 

Reviewer #1: No

Reviewer #2: Yes

3. Have the authors made all data underlying the findings in their manuscript fully available?

Reviewer #1: Yes

Reviewer #2: Yes

4. Is the manuscript presented in an intelligible fashion and written in standard English?

Reviewer #1: Yes

Reviewer #2: Yes

5. Review Comments to the Author

Reviewer #1: This article, which attempts a meta-analysis of the preventive effect of antidepressants on COV-19 infections, should be rejected for the following reasons.

1. There are already meta-analysis papers showing the efficacy of fluvoxamine for COVID-19 (PMID: 35385087, 35383578).

2. Only three RCTs could be used for the meta-analysis, although 12 articles with a variety of methods, including RCTs and case-control, observational studies, were extracted. The results are not clinically useful information because of the varied treatment settings and severity of illness of the subjects. Ultimately, this indicates that the number of papers is still insufficient for a meta-analysis on this topic.

Reviewer #2: This is a review of the manuscript “The effect of antidepressants on severity of COVID-19 in hospitalized patients: A systematic review and meta-analysis” submitted for publication in PLOS One. This is a very interesting manuscript, which addresses a timely and important research question, with potential important therapeutic implications. Below are several comments and suggestions to strengthen the submission that I hope the authors will find useful.

1/ Since the literature on this topic is progressing fastly, this meta-analysis would benefit in being actualized, until June 2022 for example. This would be particularly important since a very large observational study, including fluoxetine, has been released in March 2022 (doi: 10.1093/ofid/ofac156. PMID: 35531374).

2/ I would discuss the exclusion from the meta-analysis of studies not reporting how COVID-19 was assessed. If it was clinically or if the study is based on EHR data, it constitutes a major bias as patients may reach the outcome (e.g. death or hospitalization) for another reason.

3/ As indicated in the introduction, the severity of COVID-19 is known to be higher in patients with medical comorbidities, and people taking antidepressants, mainly because they suffer from major depression or anxiety disorders, are more likely to have medical comorbidities than their counterparts. Therefore, studies not taking into account of baseline comorbidities, indication of the treatment (i.e., depressive disorder, anxiety disorder, dementia, neurologic pain), or baseline severity in their analyses are likely to provide biased estimates, as recently shown (doi: 10.1038/s41380-021-01393-7. PMID: 34837060; doi: 10.1016/j.bpsgos.2021.12.007. PMID: 35013734).

Therefore, for observational studies and case-control studies, I would recommend to also extract data on medical comorbidities (whether analyses of each study took into account them and how), medical indication, and clinical severity at baseline, to help judge the readers the quality of each study. Studies without randomization not properly adjusting for comorbidities and age, the 2 main risk factors for severe COVID-19, cannot be considered at low risk of bias. These data may also reveal a clear source of heterogeneity of results across studies if only studies not adjusting for comorbidities do not find significant protective associations.

4/ Study period and country, allowing to approximate the dominant variant at the time of each study, may be also worth to be included, to examine whether the effect of antidepressants may apply across all variants.

5/ Importantly, I think it would be interesting and important to present in this article a metanalysis of observational studies, 1 for any antidepressant or SSRIs, and 1 for each molecule for which there are for example at least 3 different studies (e.g., fluoxetine), by including calculated effect size (expressed as SMD for example) of each study based on adjusted results (e.g. AOR). Alternatively or additionally, a meta-analysis including only matched observational data can be performed. It would give a sense of the magnitude of the effect and of the heterogeneity of the results. A global estimate of the effect of antidepressants would strongly increase the interest for this article.

6/ To understand conflicting results across studies, I would recommend performing sensitivity analyses: 1) excluding studies not reporting how COVID-19 was detected (i.e., without specifying if individuals had a positive RT-PCR or antigen test) if not already excluded (see comment #1); 2) excluding studies not taking into account baseline comorbidities as it is a central confounding factor associated with both antidepressant use and COVID-19-related outcomes, 3) excluding studies not reporting the dose of antidepressant, as these studies may have included in their analyses from EHR individuals who had a past but not a current prescription of antidepressant (providing the dose of antidepressant is a way for increasing the confidence that the patient actually received the treatment at the time of the infection in such study), and 4) keeping only truly “high-quality” / “at low-risk of bias” studies not excluded in 1), 2), and 3).

Furthermore, only the observational studies kept in 4) can be considered at low risk of bias; to the reviewer’s point of view, a study in which (i) COVID-19 status is uncertain in the absence of a specific test, (ii) compared groups are not adequately balanced in terms of medical comorbidities by an adequate adjustment, and (iii) there is uncertainty whether the patients actually took the treatment given the absence of information on the dose, cannot be considered at low risk of bias, contrary to what is indicated in L171 of the manuscript. This information should also be detailed in the result section.

A significant pooled association in the last sensitivity analysis of studies with low-risk of bias would give strength to the putative conclusion that higher-quality studies converge in finding significant protective effect of antidepressants, whereas lower-quality studies (with substantial risk of bias) find either a non-significant or a deleterious effect of antidepressants.

7/ I’m surprised by the presence of “unclear” for the Reis et al. study in the Table 4. The multi-arm RCT TOGETHER trial design seems to be of high quality, but maybe the unclear information can be found on the website https://www.togethertrial.com/trial-specifications. I tend to think that this study should be added to “the exceptions” L173.

8/ L240: The primary outcome of the Reis et al. study was “as a composite endpoint of hospitalisation defined as either retention in a COVID-19 emergency setting or transfer to tertiary hospital due to COVID-19” (the study was stopped for superiority for the primary outcome). In a separate letter (see Author's reply. Lancet Glob Health. 2022 Mar;10(3):e333), the main authors indicate that the outcome (hospitalisation or emergency care >24 h) and the outcome for FDA criterion for severe COVID-19 (used by the FDA to approve molnupiravir and Paxlovid) was also significant. In addition, the per-protocol analysis indicates a significant reduction in mortality [OR 0·09; 95% CI 0·01–0·47], and there were no differences in number of treatment emergent adverse events or drug–drug interactions. I think this information is important to be specified, before presenting secondary outcomes, for which the study was not powered. In addition, in the pooled RR for clinical trials, I think that the result of the primary outcome should be better used, and that a separate sensitivity analysis within the same Figure including the result of the per-protocol analysis for the main outcome of the Reis et al. study would give a better sense of the true effect size of fluvoxamine.

9/ L245. Similarly, the result for the main outcome of the Calusic et al study should be clearly indicated before presenting secondary outcomes, for which the study was not powered.

10/ L266: “One study proposed that antidepressants such as SSRIs and TCAs increase the risk of severe COVID-19 due to their anticholinergic effect that is likely to cause pneumonia”. This sentence should be balanced by the fact that this study (McKeigue et al.) did not specify how COVID-19 was ascertained, did not adjust for comorbidities and medical indication (to correct if true in the Table 3), and did not indicate the dose used, and, therefore, should be considered at high risk of bias.

11/ Tables 2 and 6: In the Hoertel et al. study, confounding factors (i.e., medical comorbidities, age, sex, medical indication, clinical and biological baseline severity) were identified and adjusted for. In addition, doses of each antidepressant are indicated in the Supplementary Table 1 of the article (e.g., fluoxetine 20 mg per day). Tables 2 and 6 should be corrected accordingly.

12/ L307: I suggest to revise the following sentence « This could explain the findings of Hoertel et al. study (8), although considering that there are other drugs such as chlorpromazine with ASM inhibition activity that didn’t show the same protective effects on mortality in COVID-19 patients, this mechanism is unlikely a major factor for said effect”, which sounds speculative.

Indeed, effect on ASM of a medication in a patient is likely to depend on (i) the magnitude of the in vitro effect of the ASM (https://www.ncbi.nlm.nih.gov/pmc/articles/PMC3166082/), (ii) the dose prescribed, and (iii) the concentration in the lungs (https://f1000research.com/articles/10-477).

In the cited study on chlorpromazine, the mean dose was low (e.g., 70 mg/d) and the in vitro inhibition of ASM of chlorpromazine is far lower than that of fluoxetine and fluvoxamine (https://www.ncbi.nlm.nih.gov/pmc/articles/PMC3166082/).

Rather, several studies preclinical (in vitro and ex vivo) and observational studies suggest that inhibition of acid sphingomyelinase/ceramide system plays a potentially important role and may explain both potential antiviral and anti-inflammatory effects of certain antidepressants in Covid (doi: 10.1038/s41380-021-01254-3; doi: 10.1016/j.jbc.2021.100701). Particularly, among SSRIs, the magnitude of the in vitro inhibition of ASM, which varies across molecules (eg, fluoxetine > paroxetine > fluvoxamine > other SSRIs) (doi: 10.1371/journal.pone.0023852), appears to correlate with the magnitude of the in vitro antiviral effect against SARS-CoV-2 (doi: 10.1080/22221751.2020.1829082; doi: 10.1001/jamanetworkopen.2021.36510). Furthermore, a retrospective cohort study of an adult psychiatric facility operated by the New York State Office of Mental Health found a significant and substantial protective association between the use of antidepressants, and particularly fluoxetine and trazodone, and COVID-19 infection (doi: 10.1192/bjo.2021.1053). In addition, a recent study found that patients taking a FIASMA antidepressant at baseline had a significantly reduced risk of intubation or death compared with those taking a non-FIASMA antidepressant at baseline, adjusting for sociodemographic characteristics, psychiatric and other medical comorbidity, and other medications (doi: 10.1038/s41398-022-01804-5. PMID: 35241663). Finally and importantly, four studies found that plasma ceramide levels strongly correlate with clinical and inflammation severity among patients with Covid-19 (DOI: 10.1038/s41598-021-00286-7; DOI: 10.3390/ijms221910198; DOI: 10.3390/ijms22094794; https://www.medrxiv.org/content/10.1101/2022.01.19.22269391v1). These data may enrich the discussion.

13/ To the reviewer’s point of view, S1R agonist effect as a central explanation of the effect of fluvoxamine remains uncertain in the absence of any preclinical or clinical data specific to Covid supporting this effect. More importantly, based on the results of this meta-analysis, it seems unable to explain associations observed between non-(or very low)-S1R antidepressants (such as paroxetine, mirtazapine or venlafaxine, which are FIASMAs (https://www.ncbi.nlm.nih.gov/pmc/articles/PMC3166082/), and reduced risk of intubation or death (doi: 10.1038/s41380-021-01021-4) as well as anti-inflammatory effects observed with a broad range of antidepressants (and not only the S1R antidepressants fluvoxamine and fluoxetine) in individuals with major depression (doi: 10.1007/s12035-017-0632-1).

14/ Beyond S1R and FIASMA potential mechanisms, anti-inflammatory properties of antidepressants might also be explained by non–S1R-inositol-requiring enzyme (IRE) pathways, such as nuclear factor κ B, inflammasomes, Toll-like receptor 4, or peroxisome proliferator-activated receptor γ (doi: 10.1001/jamanetworkopen.2021.36510; doi: 10.1038/s41380-021-01254-3).

15/ The terms “severity predictors” and “experimental studies” are confusing. I suggest to revise them as “outcomes” and “clinical trial” for example.

16/ Conclusion and abstract: Although fluvoxamine is the only antidepressant hat has been tested in RCTs against COVID, there is no observational evidence of superiority to other antidepressants; contrariwise, fluoxetine seems to be the most studied with the greatest effect size in observational studies. I think that the conclusion could include this point, which may suggest prioritizing fluoxetine and fluvoxamine in future clinical trials.

6. PLOS authors have the option to publish the peer review history of their article (what does this mean?). If published, this will include your full peer review and any attached files.

Reviewer #1: **Yes: **Jun-ichi Iga

Reviewer #2: No

---

## [Author Response · Author response to Decision Letter 0]

23 Aug 2022

We thank the editor and the reviewers for their comments on our manuscript. Below is our response to each point raised by the academic editor and reviewers. We hope that we satisfyingly addressed them and that the manuscript will be now suited for publication.

Sincerely,

On behalf of all authors,

Moein Zangiabadian

Dear editor

Thank you for considering our manuscript. We thank the editor and reviewers for their thoughtful critique and comments. We have carefully edited the manuscript as requested by you and have provided a point-by-point response below. Please find the revised version included. The revisions are highlighted in yellow in the resubmitted manuscript. We hope this meets the established reputation for the quality of your esteemed journal.

1) Thanks for pointing this out. The manuscript is updated with PLOS ONE's style requirements.

2) Thanks for your fair and constructive comment. The duplicate text in discussion is rephrased word by word.

3) Thanks for pointing this out. The supplementary file (excel of data extraction) for data availability statement is uploaded and the sentence of this section is changed. (Page 15: line 576)

4) Thanks for pointing this out. The ORCID iD is added in Editorial Manager.

5) Thanks for pointing this out. The tables S1&S2&S3 are uploaded as supplementary file (search strategy). 

Dear reviewer 1

We appreciate the time and attention you spent in reviewing our manuscript. 

1) Our manuscript is a systematic review of the preventive effect of antidepressants on COV-19 infections and we have discussed about different types of antidepressants in our manuscript and the meta-analysis of the effect of fluvoxamine on COVID-19 is only a part of our results not all of them. Your two mentioned studies are only about the fluvoxamine and have not discuss about other antidepressants. The first study (PMID: 35385087) has included three RCTs for meta-analysis and one of them (Lenze E. Fluvoxamine for early treatment of COVID-19: a fully-remote, randomized placebo controlled trial) was not in our analysis so we added it. On the other hand, one of our included studies (Calusic M. Safety and efficacy of fluvoxamine in COVID-19 ICU patients: An open label, prospective cohort trial with matched controls) was not in their analysis so the new analysis with five studies that we have performed, is updated in this subject. The second study (PMID: 35383578) has included three studies with different methods (two RCTs and one prospective cohort) and has analyzed them but this is not correct to analyze interventional and observational studies together. The three included articles of this study are common with our included studies. 

2) There were diversity in study designs and type of antidepressants and we only could analyze the clinical trials about the effect of fluvoxamine. So we did not restrict our results to this analysis and systematically reviewed the effect of other antidepressants in studies with different methods. Also the analysis is updated and five studies are included. 

Dear reviewer 2

 Thank you for your willingness to consider our initial manuscript “The effect of antidepressants on severity of COVID-19 in hospitalized patients: A systematic review and meta-analysis”. We have carefully considered all comments and revised and improved some parts of the original manuscript as requested. We hope this meets the established reputation for the quality of your esteemed journal. The revisions are highlighted in yellow in the resubmitted manuscript.

1) Thanks for your fair and constructive comment. The search is updated till June 14, 2022, and two further studies (RCTs) about the effect of fluvoxamine on COVID-19 are added. But your suggested article (doi: 10.1093/ofid/ofac156. PMID: 35531374) about fluoxetine didn’t meet the inclusion criteria. Because the aim of our study is to review the articles which evaluate the effect of antidepressants on reducing the severity of Covid-19 and MacFadden et al did not compare this aforementioned goal and focused on infection risk and detection of SARS-CoV-2, so this article was excluded. (Page 3: line 113)

2) Thanks for pointing this out. As it is shown in table 1, all included studies for meta-analysis have used PCR test for detection of COVID-19. So there is no concern about your mentioned bias.

3) Thanks for your recommendation. The analysis adjustment for your mentioned factors is extracted from all included studies and added in table 5 and unfortunately most studies have not adjust their results by these factors so it can cause bias and we added this point to our study limitations in discussion section (Page 13: lines 524-531). On the other hand, among included clinical trials that we have analyzed, only one of them is not randomized control trial (Calusic et al.) but this study has mentioned that there was matching between two groups in the basis of age, sex, co-morbidities, vaccination status and disease severity (see table 5) so there is no concern about this study although in the quality assessment we mentioned this study with high risk of bias. 

4) Thanks for your suggestion. Study characteristics like study period and country are in table 1.

5) Thanks for your recommendation. Unfortunately it’s not possible to Meta-analyze any antidepressant or molecule because there are not enough studies (at least three studies) in the same design to analyze them together. For example, there are two cohorts and one case control study about fluoxetine or one cohort and two case control studies about escitalopram. On the other hand, the data is common between SSRIs and SNRIs in three studies and there are not separate data for each of them for analyzing. 

6) Thanks for your suggestion. As it mentioned in above comment, we only analyzed the clinical trials and the I2 for our analysis was 0.0 so the heterogeneity is very low and there was not any conflicting results in our analysis so sensitivity analysis is not necessary.

7) Thanks for pointing this out. Table 4 is changed and quality assessment section is corrected. (Page 5: lines 176-179)

8) Thanks for your suggestion. The primary outcome is mentioned in result section as a significant difference between two groups but unfortunately the raw data about this outcome is not available for analysis and we used secondary outcome raw data for our analysis. (Page 7: lines 251-258)

9) Thanks for your recommendation. The primary or main outcome in this study was not clarified and all mentioned outcomes were secondary and statistically insignificant. On the other hand, all data in Calusic et al. study were continuous except the data about mortality that was dichotomous and we used of this outcome for analysis. 

10) Thanks for your fair and constructive comment. All of your comments are correct but as we have added in table 6 this study is adjusted for comorbidities. However,table 3 and quality assessment section is changed according to the McKeigue et al. high risk of bias and the discussion is balanced. (Page 5: lines 176-179, page 8: lines 287-290)

11) Thanks for pointing this out. Tables 2 and 6 are corrected. 

12) Thanks for your fair and constructive comment. Discussion is corrected in regard of your suggestions:

• “Revision of following sentence « This could explain the findings of Hoertel et al. study (8), although considering that there are other drugs such as chlorpromazine with ASM inhibition activity that didn’t show the same protective effects on mortality in COVID-19 patients, this mechanism is unlikely a major factor for said effect”, and 3 factors including (i) the magnitude of the in vitro effect of the ASM (ii) the dose prescribed, and (iii) the concentration in the lungs” are added. (Page 11: lines 414-436)

• “Chlorpromazine inhibition of ASM in in-vitro condition” is added. (Page 11: lines 422-426)

• “Observational studies which show that inhibition of acid sphingomyelinase/ceramide as a potentially important role” is added. (Page 10: lines 396-401)

• “Both potential antiviral and anti-inflammatory effects of certain antidepressants in Covid-19” is added. (Page 10: lines 401-403)

• “the magnitude of the SSRIs in vitro inhibition of ASM” is added.(Page 10 and 11: lines 404-407)

• “Correlate of the magnitude of the in vitro antiviral effect in severity of disease” is added. (Page 11: lines 406-407)

• “A retrospective cohort study of an adult psychiatric facility operated in the New York State Office of Mental Health” is added. (Page 11: lines 437-439)

• The comment about “FIASMA antidepressant at baseline had a significantly reduced risk of intubation or death compared with those taking a non-FIASMA” is added. (Page 11: lines 439-442)

• “Four studies found that plasma ceramide levels strongly correlate with clinical and inflammation severity among these studies” are added (Page 10: lines 369-378)

13) Thanks for your fair and constructive comment. Discussion is corrected in regard of your suggestion. (Page 12: lines 468-476)

14) Thanks for pointing this out. Your suggested mechanism is added. (Page 12: lines 479-489)

15) Thanks for your recommendation. These two terms were replaced by your suggested terms. 

16) Thanks for your suggestion. Conclusion is corrected. (Page 14: L545-548, page 15: lines 557-571)

Reviewer 2 supplementary comments

1) Thanks for your suggestion. Abbreviations are removed from abstract.

2) Thanks for pointing this out. Our included studies were in three study designs (clinical trials, case controls and cohorts) so we were inevitable to use three different tools for assessing their quality. But as we have mentioned in quality assessment section, among 12 included studies only three of them had high risk of bias so the total quality of included studies is appropriate.

3) Thanks for your fair and constructive comment. Your mentioned values are added into analysis results. (Page 7: lines 273-275)

4) Thanks for pointing this out. The I2 was lower than 50% so we used fixed model. The statistically analysis section of method part is corrected and this value is added. (Page 4: line 155)

5) Thanks for pointing this out. As its not correct to mix apples and oranges in meta-analysis we didn’t perform analysis for other drugs in different study types and only analyzed the effect of fluvoxamine in clinical trials.

---

## [Decision Letter · Decision Letter 1]

21 Sep 2022

The effect of antidepressants on the severity of COVID-19 in hospitalized patients:

A systematic review and meta-analysis

PONE-D-22-10307R1

Dear Dr. ZANGIABADIAN,

We’re pleased to inform you that your manuscript has been judged scientifically suitable for publication and will be formally accepted for publication once it meets all outstanding technical requirements.

Kind regards,

Academic Editor

PLOS ONE

Additional Editor Comments (optional):

Reviewers' comments:

Reviewer's Responses to Questions

**Comments to the Author**

1. If the authors have adequately addressed your comments raised in a previous round of review and you feel that this manuscript is now acceptable for publication, you may indicate that here to bypass the “Comments to the Author” section, enter your conflict of interest statement in the “Confidential to Editor” section, and submit your "Accept" recommendation.

Reviewer #1: All comments have been addressed

Reviewer #2: All comments have been addressed

2. Is the manuscript technically sound, and do the data support the conclusions?

Reviewer #1: Yes

Reviewer #2: Yes

3. Has the statistical analysis been performed appropriately and rigorously? 

Reviewer #1: Yes

Reviewer #2: Yes

4. Have the authors made all data underlying the findings in their manuscript fully available?

Reviewer #1: Yes

Reviewer #2: Yes

5. Is the manuscript presented in an intelligible fashion and written in standard English?

Reviewer #1: Yes

Reviewer #2: Yes

6. Review Comments to the Author

Reviewer #1: The authors successfully incorporated the reviewer's comments into the revised manuscript. This paper is suitable for PLOS One in the current form.

Reviewer #2: I thank the authors for their responses. The manuscript has been importantly improved.

A minor additional comment, that should not delay the publication of this important article, would be to shortly discuss the recently published RCT in the NEJM (doi:10.1056/NEJMoa2201662), that found that fluvoxamine prescribed at a low dose (100 mg/d ) to over-weighted and obese outpatients with COVID-19 showed no significant benefit on the risk of emergency department visits, hospitalizations or death, contrasting with the findings of the RCTs TOGETHER and STOP-COVID, in which fluvoxamine was prescribed at higher doses, i.e., 200 and 300 mg/d, respectively, and with results from this meta-analysis.

This discrepancy might be explained by a potential effect of fluvoxamine occurring at a minimum dose of 200 mg/d, as suggested by a recently published observational study (doi: 10.1038/s41398-022-02109-3) that found that exposure to antidepressants, especially those with FIASMA properties, was associated with reduced incidence of emergency department visitation or hospital admission among SARS-CoV-2 positive outpatients, in a dose-dependent manner and from daily doses of at least 20 mg fluoxetine equivalents. Altogether, these findings might suggest that fluvoxamine or FIASMA antidepressants should be prescribed at a minimum dose of 200 mg/d (100 mg twice daily) to possibly observe a benefit in patients with COVID-19.

7. PLOS authors have the option to publish the peer review history of their article (what does this mean?). If published, this will include your full peer review and any attached files.

Reviewer #1: **Yes: **Jun-ichi Iga

Reviewer #2: No

---

## [Editor Report · Acceptance letter]

26 Sep 2022

PONE-D-22-10307R1 

The effect of antidepressants on the severity of COVID-19 in hospitalized patients: A systematic review and meta-analysis 

Dear Dr. Zangiabadian:

I'm pleased to inform you that your manuscript has been deemed suitable for publication in PLOS ONE. Congratulations! Your manuscript is now with our production department. 

Kind regards, 

on behalf of

Dr. Robert Jeenchen Chen 

Academic Editor

PLOS ONE